# Polyploidy as an Adaptation against Loss of Heterozygosity in Cancer

**DOI:** 10.3390/ijms23158528

**Published:** 2022-08-01

**Authors:** Marco Archetti

**Affiliations:** Department of Biology, Pennsylvania State University, University Park, PA 16802, USA; mua972@psu.edu

**Keywords:** polyploidy, heterozygosity, complementation, endomitosis, cancer

## Abstract

Polyploidy is common in cancer cells and has implications for tumor progression and resistance to therapies, but it is unclear whether it is an adaptation of the tumor or the non-adaptive effect of genomic instability. I discuss the possibility that polyploidy reduces the deleterious effects of loss of heterozygosity, which arises as a consequence of mitotic recombination, and which in diploid cells leads instead to the rapid loss of complementation of recessive deleterious mutations. I use computational predictions of loss of heterozygosity to show that a population of diploid cells dividing by mitosis with recombination can be easily invaded by mutant polyploid cells or cells that divide by endomitosis, which reduces loss of complementation, or by mutant cells that occasionally fuse, which restores heterozygosity. A similar selective advantage of polyploidy has been shown for the evolution of different types of asexual reproduction in nature. This provides an adaptive explanation for cyclical ploidy, mitotic slippage and cell fusion in cancer cells.

## 1. Introduction

Polyploidy plays an important role in speciation and evolution, but also in development and adult animal tissues [1,2]; polyploidy is so common in tumors [3] that abnormal nuclear morphology is a diagnostic criterion for cancer [4,5]. In spite of their diagnostic importance, polyploid cells were traditionally neglected in cancer research because considered senescent and unable to divide, a view that has recently changed [6]: polyploid cells can in fact divide [7,8], differentiate [9] and bud into a depolyploidized progeny [10,11,12,13] that can be more aggressive [14,15] and resistant to therapies [16,17,18,19,20,21,22,23,24,25]. While the cytological and molecular characteristics of polyploidy are known [26,27], the role of polyploidy in tumor progression and resistance to therapies is not fully understood [28,29]. What is, if any, the adaptive value of polyploidy in cancer? 

One hint comes from the fact that, remarkably, while populations of cancer cells are generally assumed to be clonal, polyploid cancer cells divide in ways that differ from standard mitosis and lead to heterogeneity in their progeny. In some cases, polyploid cancer cells fuse [30] and then divide in a kind of parasexual recombination similar to the life cycles of some protists and fungi, leading to a heterogeneous progeny of cells. In other cases, polyploid cancer cells divide by alternative mechanisms [11,12,13,28,29] such as mitotic slippage [31], the incomplete mitosis that results in a doubled genome in interphase, analogous to endomitosis [32]. Furthermore, cancer cells often express meiotic genes that are normally only expressed in germ cells [33,34,35,36,37,38,39,40,41,42] that lead, for example, to high rates of recombination [43], which is instead rare in normal mitosis. The expression of these “germ cell cancer genes” leads to a poor prognosis and resistance to therapies and their role is poorly understood [26,37,44,45].

The rationale of this paper is that polyploidy can be explained as an adaptation of cancer cells against the loss of heterozygosity (LOH) arising from these atypical types of mitosis. The same concept was used in evolutionary biology to explain the stability of different types of asexual reproduction in nature [46,47,48,49,50]. As is well-known since Stern’s pioneering experiments on somatic recombination in Drosophila [51], recombination in mitosis leads to immediate LOH for all the loci distal to the site of crossing over in half of the cell progeny (Figure 1). LOH leads to the unmasking of recessive deleterious alleles (loss of complementation), which has deleterious effects for the cell. The effect of LOH depends on the number and effect of such recessive mutations, that is, on the number of lethal equivalents (the number of recessive mutation whose combined effect is lethal when made homozygous). LOH will increase over time, and unless the number of lethal equivalents or recombination rates are low, will lead to a substantial decline in fitness strong enough to explain why sexual reproduction in nature is stable against asexual mutants (which, in the absence of LOH, have a twofold advantage against sexual individuals) [46,48]. 

Polyploidy, however, slows down the LOH as additional alleles preserve heterozygosity. Heterozygosity can also be restored by occasional amphimixis and outcrossing. The logic of this paper is that similar processes occur in cancer, where mitotic recombination leads to LOH, and complementation can be restored by cell fusion followed by multiple divisions to return to the diploid state; or can be slowed down and prevented by polyploidy or mitotic slippage, a form of somatic endomitosis in which an additional duplication at the interphase is followed by two divisions. My purpose here is to provide a quantitative prediction of this effect to understand whether selection against LOH is strong enough to provide an adaptive advantage to polyploidy, endomitosis or cell fusion.

## 2. Results

### 2.1. Loss of Heterozygosity in Mitosis with Recombination

While mitosis without recombination produces identical clones (Figure 2A), recombination leads to heterogeneous daughter cells: with one crossing-over event, when recombinant chromatids segregate apart (*x* segregation (Figure 1)), complementation is lost immediately at all loci distal to the site of crossing over (Figure 2B). LOH increases over time, but as it is counterbalanced by selection, it reaches an equilibrium after a few divisions (Figure 2C). With random segregation (*x* = 0.5) there is always a 50% probability that the daughter cell is produced by *z* segregation and, therefore, has no LOH. Additionally, if, as I assume here, only reciprocal chiasmata occur, this remains the case for multiple crossing-over events, because with reciprocal chiasmata two chromatids are always remain non-recombinant. Hence, the fitness of mitotic cells with recombination cannot be lower than 0.5, irrespective of the number of lethal equivalents and crossing-over events (Figure 2D). This is not the case, however, if *x* segregation is more frequent than *z* segregation (*x* > 0.5) (Figure 2E), as LOH increases with the probability of *x* segregation (Figure 2F). In any case, fitness reduction is so high that, even with modest frequencies of recombination, occasional spontaneous cell fusion (which immediately restores heterozygosity) has a clear advantage.

### 2.2. Polyploidy Slows down Loss of Heterozygosity

While cell fusion can restore heterozygosity after LOH has occurred in diploid cells, if cells are already polyploid, LOH would arguably be slower. Indeed, the results show that mitotic recombination in tetraploid cells leads to LOH (Figure 3A) as in diploid cells but at a lower rate. Therefore, tetraploidy confers higher fitness to cells with mitotic recombination (Figure 3B) unless the frequency of recombination is very high (*r* = 3) and the number of lethal equivalents is high (*λ* > 2.5). Note that LOH is lower with two crossing-over events than with one. This is because a second crossing-over event restores complementation to the distal part of the chromosome that had been recombined by the first crossing-over event (a third crossing-over event, however, nullifies the effect of the second event, and so on). This effect is more evident for low *λ*, also occurring, but less evident, in diploid cells (see Figure 2).

### 2.3. Endomitosis Enables a Lower Loss of Heterozygosity Than Mitosis

With endomitosis, a further division at interphase produces a transient tetraploid cell, which then divides twice to produce four diploid daughter cells (Figure 4A–C). Different patterns of LOH are produced depending on whether sister chromosomes (originated from the same chromosome at interphase) pair with each other or with a non-sister chromosome (one copy of the homologous chromosome). The pairing of sister chromosomes leads to no LOH irrespective of recombination, because even if a crossing over occurs, the parts that are exchanged are identical (Figure 4A). Pairing of non-sister chromosomes leads to total LOH on half of the daughter cells when there is no recombination (Figure 4B) and partial LOH if recombination occurs (Figure 4C). As a result, assuming random chromosome pairing, LOH increases with recombination rates and with the number of lethal equivalents (Figure 4D) but is always lower than in diploid mitosis (Figure 4E). Endomitosis, therefore, has an adaptive advantage in the presence of recombination. This advantage increases with the probability (*s*) of sister chromosome pairing (Figure 4E).

## 3. Discussion

Mitosis with recombination leads to immediate LOH at all loci distal to the site of crossing-over events in cells produced by *x* segregation, and LOH increases rapidly over time, reaching a mutation/selection balance that creates a substantial fitness cost. Spontaneous cell fusion (with different cells) confers an advantage to these cells because it immediately restores complementation of recessive deleterious mutations. This is similar to the advantage of amphimixis (sexual reproduction) against asexual reproduction in nature [46]. It is worth noting that diploid mitotic recombination is genetically equivalent to apomixis (a meiosis in which one division is suppressed) with suppression of the first meiotic division, which is more common in nature than the suppression of the second division; and that meiotic apomixis in nature is often associated with triploidy or tetraploidy, which lead to lower LOH [46]. The effect of polyploidy is strong enough that, while diploid apomixis is generally unstable against sexual reproduction, triploidy or tetraploidy enable apomictic mutants to invade and drive to extinction sexual populations [46,48]. 

The same effect of polyploidy is observed here for mitotic recombination: tetraploidy leads to lower LOH than in diploid cells; therefore, a tetraploid mutant cell would have a proliferation advantage within a tumor against its diploid neighboring cells. One important difference is that the fitness of cancer cells with LOH and polyploidy is relative to other cancer cells with LOH, whereas in studies on the evolution of asexual reproduction in multicellular organisms, polyploid individuals with LOH compete with individuals that reproduce sexually and have no LOH. Selective pressure for polyploidy is therefore weaker in tumors. On the other hand, selection against LOH is so strong in apomictic populations that asexual mutants can become extinct before polyploidy emerges in a new asexual lineage [46]. Therefore, polyploidy against LOH seems more likely to evolve in tumors than in a whole-organism situation, in spite of the weaker selection.

Endomitosis is an alternative way to prevent LOH that does not require a permanent increase in ploidy level but only a transient tetraploidization at interphase, followed by two divisions. Endomitosis with pairing of sister chromosomes leads to no LOH irrespective of recombination, whereas the pairing of non-sister chromosomes leads to total LOH in half of the daughter cells when there is no recombination and partial LOH if recombination occurs. The combined effect is, as we have seen, a lower LOH than with diploid mitosis. Hence, mitotic slippage (somatic endomitosis) is also an adaptation against LOH in cancer cells with high rates of mitotic recombination. Again, a similar effect is observed for endomitosis in natural populations [46,47,48,49,50].

The main overall result is that LOH leads to a selective pressure for the evolution of polyploidy, cell fusion or endomitosis due to mitotic recombination. The analysis presented here assumes that only reciprocal chiasmata occur, which is not necessarily true; however, because the effect of non-reciprocal chiasmata is only relevant if there is more than one crossing-over event per duplication, which is likely uncommon, neglecting non-reciprocal chiasmata is unlikely to be an important assumption. Moreover, non-reciprocal chiasmata would arguably only increase LOH, as multiple recombination events in the same chromatid pairs distal to the first chiasma can only reduce LOH for that pair. A more important assumption is that LOH is deleterious because it unmasks recessive deleterious mutations.

On the contrary, it could be argued that LOH may be beneficial to the tumor in some cases, for example by allowing normally recessive proto-oncogenes to be expressed, or by allowing a cell to become homozygous for a mutation that disables a tumor-suppressing gene. It has been known for a long time, for example, that LOH is a key event in tumorigenesis for retinoblastoma [52,53,54]. It is also possible that a heterogeneous tumor comprised of different homozygous clones has a proliferation advantage over a homogeneous heterozygous tumor, even for genes that are essential for tumor progression, such as growth factors, because heterogeneous cell populations can grow faster [55]. Intra-tumor heterogeneity is common [56,57], and it has implications for cancer progression, diagnosis, and treatment [58,59,60] but remains poorly understood [61]. LOH can help explain the origin of heterogeneous homozygous clones from a homogeneous heterozygote population. In this case, polyploidy would not be necessary to explain the maintenance of heterozygosity.

However, LOH at a few specific loci is different from generalized LOH produced by mitotic recombination, which entails a loss of complementation for thousands of genes distal to the site of crossing over, including numerous non-driver genes that are essential for the cell [62,63,64], and is more likely to be overall deleterious for cells. In tumors, gene expression is higher for genes conserved with unicellular organisms, which have generally systemic effects (such as housekeeping genes) than for genes of metazoan origin [65,66], especially in polyploid cancer cells [67]; therefore, the coordinated expression of interacting multicellularity and unicellularity processes is lost in cancer progression, and LOH may enhance this mismatch. It has been suggested [68] that the unmasking of recessive deleterious alleles may not be critical because the compensatory growth of the remaining viable cells allows that clonal lineage to thrive anyway. While this may be the case, the relevant point here is that, if it unmasks recessive deleterious mutations, LOH leads to a reduction of fitness; hence, it creates a selective pressure for the evolution of polyploid or endomitotic cells (which are protected from the deleterious effects of LOH) or for spontaneous cell fusion followed by ploidy levels returning to normal (which restores LOH).

As we have seen, in spite of the adaptive advantage conferred by polyploidy, endomitosis or cell fusion, in the presence of mitotic recombination, LOH remains. Indeed, LOH is common in cancer development, and in certain tumor types it can affect more than 20% of the genome [69]. As LOH creates genetic differences between tumor and normal cells, it provides opportunities for development of biomarkers and vulnerabilities that can be exploited to develop therapies [70,71,72]. These strategies should take into account the fact that polyploidy will likely evolve whenever LOH occurs; therefore, it will interfere with any such diagnostic and therapeutic interventions against LOH.

## 4. Methods

I assume that a chromosome has 1000 heterozygous diploid loci; for tetraploid cells, I assume that polyploidy is due to whole-genome duplication; hence, there are two copies of two different alleles. At each generation:The number of homozygous loci is measured.Fitness is calculated as 1-*h*·*λ*, where *h* is the fraction of homozygous loci and *λ* is the number of lethal equivalents (one lethal equivalent is one recessive mutation whose effect—or more mutations whose summed effects—is lethal when made homozygous); if *h*·*λ* > 1, fitness is equal to 0 (because an individual can have more than 1 lethal equivalent but cannot die more than once).A cell is chosen to reproduce with a probability proportional to its fitness.For endomitosis only: chromosomes are duplicated and paired; pairing occurs between sister chromosomes with probability *s* or between non-sister chromosomes with probability 1-*s*.If 0 < *r* < 1, a locus *p* and two non-sister chromatids are chosen at random, and recombination occurs with probability *r*: the alleles at all loci distal to the crossing over position *p* are swapped between the two chromatids (therefore, I assume that crossing over always leads to recombination rather than just gene conversion). If *r* > 1 (in this case, *r* is an integer), a locus *p* and two non-sister chromatids are chosen at random, and recombination occurs with probability 1: the alleles at all loci distal to the crossing-over position *p* are swapped between the two chromatids; the process is repeated for a total of *r* times choosing a different locus (no chromosome interference) but with the same chromatids (chiasmata are reciprocal).Segregation of chromatids occurs: for mitosis, *x* segregation (in which each recombinant chromatid segregates with the non-sister non-recombinant chromatid) occurs with probability *x*, and *z* segregation (in which the two recombinant chromatids segregate together, apart from the two non-recombinant chromatids) occurs with probability 1-*x*.One random daughter cell produced by this process is chosen to replace the original cell.The process 1–7 is repeated for 1000 cells.The process 1–8 is repeated for 200 generations or until a stable value of LOH is reached (that is, the average over 50 generations does not change by more than 1%).The average value of LOH and fitness is measured as the average over the last 50 generations.The process 1–10 is repeated 10 times, and the results are averaged.

## Figures and Tables

**Figure 1 ijms-23-08528-f001:**
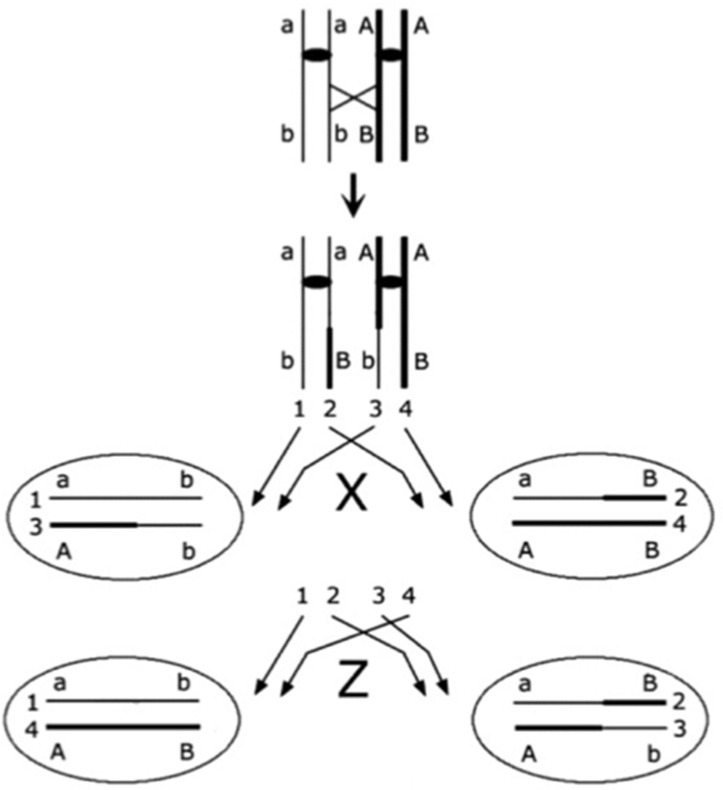
A schematic representation of mitosis with crossing over. Each segment represents a chromatid, with sister chromatids linked by a centromere (black oval); crossing over (diagonal cross) occurs between the centromere and locus B. Two types of segregation are possible (Stern [51]): *z* segregation, in which the two recombinant chromatids (2 and 3) segregate together, apart from the two non-recombinant chromatids (1 and 4); and *x* segregation, in which each recombinant chromatid segregates with the non-sister non-recombinant chromatid. Loss of heterozygosity occurs for all loci distal to the site of crossing over with *x* segregation (in this example, at locus B).

**Figure 2 ijms-23-08528-f002:**
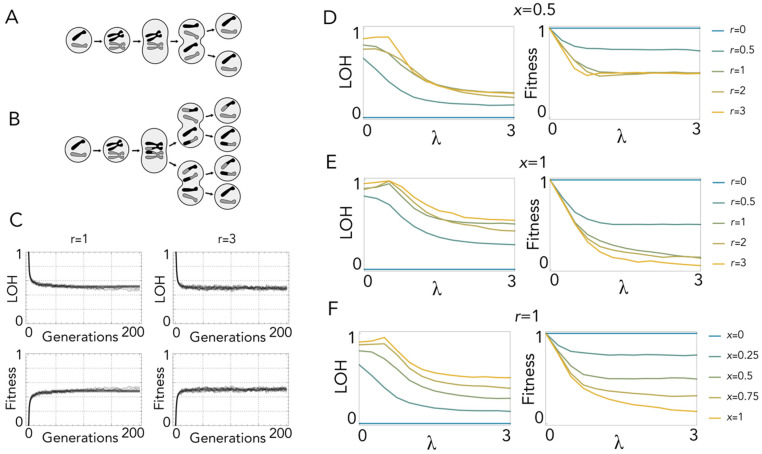
Loss of heterozygosity in mitosis. (**A**): Mitosis without recombination: there is no LOH. (**B)**: Mitosis with one crossing-over event: daughter cells produced by *x* segregation have LOH. (**C**): Values of LOH and fitness over 200 generations with one lethal equivalent (*λ* = 1) and one (*r* = 1) or three (*r* = 3) crossing-over events with random segregation (probability of *x* segregation *x* = 0.5). (**D**): LOH and fitness at equilibrium as a function of *λ* (the number of lethal equivalents) for different values of *r* (number of crossing-over events per replication) with *x* segregation occurring with the same probability (*x* = 0.5) as *z* segregation. (**E**): LOH and fitness at equilibrium as a function of *λ* for different values of *r* with *x* segregation only (*x* = 1). (**F**): LOH and fitness at equilibrium as a function of *λ* for different values of *x* with one crossing-over event (*r* = 1).

**Figure 3 ijms-23-08528-f003:**
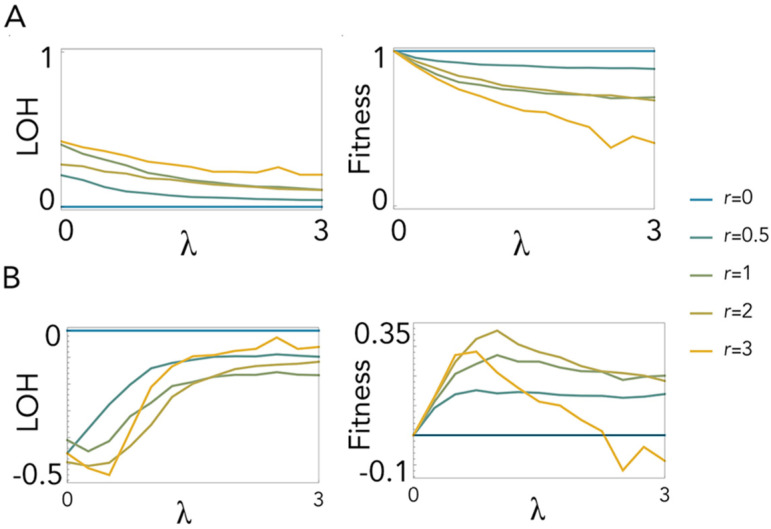
Loss of heterozygosity in mitosis with tetraploidy. (**A**): LOH and fitness at equilibrium. (**B**): Differences in LOH and fitness between tetraploid and diploid mitosis. LOH and fitness (for **A**) or their difference (for **B**) are plotted as a function of *λ* (the number of lethal equivalents) for different values of *r* (number of crossing-over event per replication) with *x* segregation occurring with the same probability (*x* = 0.5) as *z* segregation.

**Figure 4 ijms-23-08528-f004:**
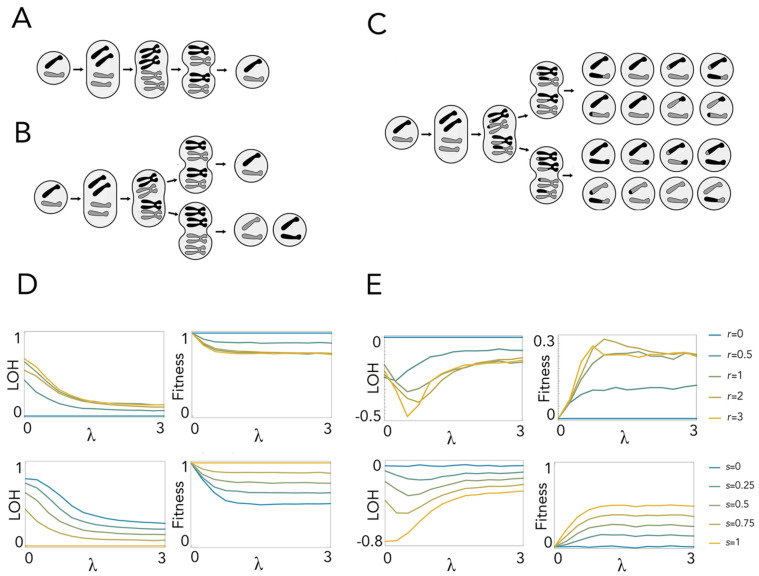
Loss of complementation in endomitosis. (**A**): Endomitosis with pairing of sister chromosomes: there is no LOH, with or without (shown here) recombination. (**B**): Endomitosis with pairing of non-sister chromosomes and no recombination: total LOH occurs in half of the daughter cells. (**C**): Endomitosis with pairing of non-sister chromosomes and recombination (one crossing-over event): partial LOH occurs in some of the daughter cells. (**D**): LOH and fitness at equilibrium for endomitosis. (**E**): Differences between endomitosis and diploid mitosis. LOH and fitness (for **D**) or their difference (for **E**) are plotted as a function of *λ* (the number of lethal equivalents) for different values of *r* (number of crossing-over events) with *s* = 0.5 (random pairing of chromosomes) and for different values of *s* (the probability of sister chromosome pairing) with *r* = 1 (one crossing-over event).

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
