# Peer review of "Polyploidy as an Adaptation against Loss of Heterozygosity in Cancer"

_ijms, 2022, doi:10.3390/ijms23158528_

Round 1
Reviewer 1 Report
The author raises an important topic of the deleterious nature of loss of heterozygosity (LOH), and the role of polyploidization in cancer cells for prevention of this process. The obtained data are novel, interesting, and scientifically sound. I believe that this paper can be published in IJMS after certain textual improvements (minor revision).
The author discusses the limitations of his approach, which is highly commendable. However, my main concern is with a possible limitation that was not discussed. The author considers cancer cells as if they were completely unicellular (UC). Then, LOH of any genes would decrease cell fitness. Yet, they are cells of multicellular (MC) organism, albeit with enhanced UC properties.
It was shown on the single-cell transcriptomes, with control of cell cycle activity, that human genes of UC origin and of UC giant cluster of interactome are stronger expressed in cancer cells compared with normal cells, and in invasive cancer cells compared with non-invasive cancer cells (Vinogradov, Anatskaya, 2020, FEBS J, 287: 4427-4439). In polyploid cancer cells, the UC genes and UC giant cluster of interactome are activated further, as compared with diploid cancer cells (Anatskaya, Vinogradov, 2022, IJMS 23: 3542). Thus, we can see that MC control over UC genes is attenuated gradually in the course of cancer progression. The author assumes that LOH (unsheltering of deleterious alleles that results in poorer gene functioning) of any genes in cancer cells reduces their fitness. However, the poorer functioning of MC-control genes would further unleash UC genes in cancer cells, thus increasing their fitness in cell competition in cancer tissues. In my opinion, this possibility should be discussed.
Smaller points:
1) It would be nice to discuss an analogy with the whole-organisms situation (adaptive nature of chromosome reshuffling and crossover) in some more details, because it is interesting to form an integrative viewpoint on cellular and organismal data.
2) The author said, "One reason is that recombination itself confers advantages, in spite of the deleterious effects of LOH, for example because it enables the repair of double stranded breaks, which may be more common in cancer cells than in normal cells."
The "may be" here is not accurate. The higher frequency of double-strand breaks in cancer cells is well-established (e.g., papers on 'genome chaos').
3) In the Abstract, the author says, "I", then in the text, "we". It looks somewhat inconsistently.
Reviewer 2 Report
The manuscript is interesting and describes a relatively unexplored but relevant idea in cancer research. The result that polyploidy can be the result of adaption to LOH is intriguing.
My only two comments are one critique and one question. For the first one: the model is not described in any detail which makes my review provisional upon seeing how it was implemented and how much would it support the interesting results presented. The comment is about intra tumor heterogeneity which I'd wonder whether it could be discussed in the discussion section as clearly LOH and polyploidy impact it but also ITH drives key aspects of tumor evolution and would impact adaptation .
Round 2
Reviewer 2 Report
I have not found any issues with the current version of the manuscript